# Characterization of Homeodomain Proteins at the Aβ Sublocus in *Schizophyllum commune* and Their Role in Sexual Compatibility and Development

**DOI:** 10.3390/jof11060451

**Published:** 2025-06-13

**Authors:** Chen Chu, Dongxu Li, Changhong Liu

**Affiliations:** State Key Laboratory of Pharmaceutical Biotechnology, School of Life Sciences, Nanjing University, Nanjing 210023, China; mg1930112@smail.nju.edu.cn (C.C.); laolang_2012@163.com (D.L.)

**Keywords:** fungal genetics, Aβ mating-type locus, homeodomain, protein interaction, sexual compatibility regulation

## Abstract

The A mating-type locus in *Schizophyllum commune*, which encodes homeodomain (HD) transcription factors, is essential for regulating sexual compatibility and development. While the role of the Aα sublocus and its Y-Z HD protein complex is adequality understood, the function of HD proteins at the Aβ sublocus remains unclear. In this study, we analyzed the Aβ sublocus of eight monokaryotic *S. commune* strains derived from the parental dikaryotic strain 20R-7-Z01 and identified four HD genes, *abr*, *abs*, *abv*, and *abq*, located at the Aβ sublocus. These genes encode two HD1 proteins (S and Q) and two HD2 proteins (R and V). Protein structure prediction, interaction assays, and in vivo functional analyses revealed that R-S and V-Q interactions independently regulate sexual compatibility and fruiting body development. This research highlights the critical role of the Aβ sublocus in fungal reproduction and provides valuable insights for breeding edible and medicinal *S. commune* strains.

## 1. Introduction

The tetrapolar fungus *Schizophyllum commune* is a typical white-rot fungus belonging to the genus *Schizophyllum*, family Schizophyllaceae, order Agaricales, class Agaricomycetes, and phylum Basidiomycota [1]. As a model organism of the genus *Schizophyllum*, it has been extensively studied in fungal genetics and developmental biology [2]. Additionally, *S. commune* is a rare edible and medicinal mushroom with high nutritional and pharmacological value [3,4,5]. It produces various valuable metabolites, including schizophyllan, which exhibits immunomodulation, antitumor activity, and wound-healing properties [6]. The bioactive compounds possess antioxidant and anti-inflammatory activities by neutralizing free radicals and reducing oxidative stress [7]. Its fermentation products are widely used as additives in the food and feed industries, underscoring its substantial industrial value [8]. However, if used improperly, *S. commune* may lead to pathogenic outcomes: it can infect fruit trees and cause plant diseases [9], and inhalation of its spores may cause respiratory infections in humans [10]. Therefore, developing compatible or improved strains of *S. commune* is essential for safe and beneficial applications.

Sexual reproduction in *S. commune* is controlled by two unlinked mating-type loci, A and B, located on different chromosomes [2]. To simplify the expression and identification of mating types, Raper et al. introduced the symbols “x” and “y” to represent the two distinct sets of A and B loci in dikaryotic strains, resulting in the genotype AxAyBxBy [11]. During sexual reproduction, four potential mating genotypes can arise: AxBx, AxBy, AyBx, and AyBy, with the compatible mating combinations being AxBx × AyBy or AxBy × AyBx. These combinations give rise to four types of mating interactions: (1) A ≠ B ≠ (Compatible), (2) A = B = (Common-AB), (3) A ≠ B = (Common-B), and (4) A = B ≠ (Common-A) [12].

The A mating-type locus is primarily involved in the nuclear pairing of two monokaryotic strains and the initiation of clamp cell formation during sexual compatibility [13,14]. This locus consists of two linked subloci, Aα and Aβ, each containing numerous alleles. Through extensive sampling, Raper et al. labeled mating-able subloci with distinct specificities, estimating that Aα in *S. commune* has approximately nine specificities (e.g., Aα1, Aα2, Aα3...Aα9), while Aβ harbors around 32 specificities (e.g., Aβ1, Aβ2, Aβ3...Aβ32) [15]. These sublocus-specific numbers were later adopted for the numbering of alleles. The combination of these specificities determines the mating type of monokaryotic individuals. Successful sexual compatibility requires at least one differing specificity in the Aα and/or Aβ loci between two monokaryotic strains. For example, two strains with identical Aα3-specific loci cannot regulate sexual compatibility, whereas one strain with Aα3 specificity and another with Aα4 specificity can activate the mating pathway. The same principle applies to the Aβ sublocus.

At the genetic level, the A locus encodes homeodomain (HD) transcription factors, which are classified into two evolutionarily related groups, HD1 and HD2, based on structural homology and sequence variation in their DNA-binding motifs [16,17]. HD2 proteins exhibit greater sequence conservation than HD1 proteins [18]. Both Aα and Aβ subloci are polygenic, with multiple genes contributing to mating-type specificity [2,18]. The genomic region surrounding the Aα sublocus in *S. commune* is highly conserved, and this locus is situated between the mitochondrial intermediate peptidase (*mip*) gene and beta-flanking (*β-fg*) gene. While Aβ is linked to Aα, its location is typically confirmed by identifying HD genes located upstream and downstream of the Aα sublocus [19,20,21].

The Aα sublocus encodes a pair of HD proteins, Y (HD2) and Z (HD1), which interact to regulate sexual compatibility [22]. The Y and Z proteins have distinct specificity numbers matching those of the Aα sublocus (e.g., the Aα4-specific sublocus encodes the Y4 and Z4 proteins, corresponding to the encoding genes *aay4* and *aaz4*). The gene structures of several Aα-specific subloci, including Aα1, Aα3, Aα4, and Aα5, have been well-characterized [18,23]. Notably, the Aα1-specific sublocus is unique in encoding only a single Y1 protein [24]. When proteins from different Aα-specific subloci, such as Y3 and Z4 or Y1 and Z3, coexist within the same cell, they form heteromultimers that activate the sexual compatibility pathway. However, identical Y and Z proteins (e.g., Y3 and Z3 or Y4 and Z4) do not activate this pathway; this interaction mode is known as non-self-recognition or non-self-combinatorial interaction [16,23]. These interactions at the Aα sublocus are adequality understood and serve as a model for the study of fungal mating and sexual reproduction.

In contrast, the role of the Aβ sublocus in *S. commune* remains less well characterized. Early hypotheses have suggested that the Aβ sublocus might function similarly to Aα, though conclusive evidence is lacking due to the absence of clearly identified HD genes in the Aβ region. Shen et al. reported an HD gene associated with Aβ6 specificity, termed *abv6*, encoding the V6 protein [25]. Additionally, genomic annotations by Ohm et al. predicted six HD genes within the Aβ sublocus of *S. commune* H4-8: *abq6* (Q6, HD1), *abr6* (R6, HD2), *abs6* (S6, HD1), *abt6* (T6, HD1), *abu6* (U6, HD1), and *abv6* (V6, HD2) [2]. Since Ohm et al. utilized the NCBI nonredundant set and Gene Ontology databases for gene annotation, it is possible that the specificity of the H4-8 Aβ sublocus might differ from that of Aβ6. Nevertheless, this did not prevent the analysis of the gene structure of the Aβ sublocus. However, these studies did not examine the expression or functional roles of these proteins during mating compatibility. Consequently, it remains unclear whether the HD proteins encoded by the Aβ sublocus contribute to sexual compatibility and fruiting body development. However, based on the reported gene structure of the Aβ sublocus, it is apparent that, similar to the Aα sublocus, the Aβ region may also harbor multiple HD1-HD2 gene pairs, contributing to a more complex network of gene interactions regulating sexual compatibility.

Investigating the Aβ sublocus offers a more refined model for elucidating the mechanisms of sexual reproduction in fungi and provides valuable insights for breeding and functional gene discovery in edible and medicinal mushrooms. Once the genetic structure of the Aβ sublocus is characterized, molecular markers can be employed to screen for desirable traits, and targeted modifications may be achieved through gene editing technologies. Therefore, in this study, we aim to address a critical gap in the understanding of the molecular mechanisms underlying sexual reproduction in *S. commune*, focusing on the poorly characterized Aβ sublocus, which may be essential for genetic improvement strategies.

## 2. Materials and Methods

### 2.1. Strains, Plasmids, and Culture Conditions

The *S. commune* strains used in this study are detailed in Appendix A. For in vivo functional validation, Bimolecular Fluorescence Complementation (BiFC) strains and transfected strains were generated using the wild-type strains 20R-7-F01-S4y (AxBx) and 20R-7-F01-S5y (AyBy). All of the wild-type strains, including 20R-7-F01-S4y (AxBx), 20R-7-F01-S17 (AxBx), 20R-7-F01-S5y (AyBy), 20R-7-F01-S192 (AyBy), 20R-7-F01-S103 (AxBy), 20R-7-F01-S180 (AxBy), 20R-7-F01-S206 (AyBx), and 20R-7-F01-S10 (AyBx), were monokaryotic strains obtained through sexual reproduction of the dikaryotic parent strain 20R-7-ZF01. These eight monokaryotic strains constituted a complete set of mating subtype strains, representing the full range of monokaryotic strains isolated from strain 20R-7-ZF01 [26]. 20R-7-ZF01 was isolated from 1966 mbsf sediments during International Ocean Discovery Program (IODP) Expedition 337 [27]. Strains were cultivated aerobically on modified minimal medium (MM) consisting of 0.046% (*w*/*v*) KH_2_PO_4_, 0.1% (*w*/*v*) K_2_HPO_4_, 0.05% (*w*/*v*) MgSO_4_, 0.15% (*w*/*v*) L-Asparagine, 2% (*w*/*v*) glucose, and 2% (*w*/*v*) agar. All cultures were maintained at 30 °C under aerobic conditions and stored at 4 °C.

*Escherichia coli* strains XL1-Blue MRF’ [28] and BL21 (DE3) [29] were used for plasmid construction, cloning, and protein expression. These strains were grown in liquid Luria–Bertani (LB) medium at 37 °C with shaking at 180 rpm [30]. Plasmids pIG1783 (ampicillin resistance) [31] and plasmid pET-28a^(+)^ (kanamycin resistance) [32] were utilized as cloning and expression vectors, respectively. Protein expression was performed in *E. coli* BL21 (DE3) strains. Protein expression was induced at an OD_600_ of 0.4–0.8 using 0.5 mM isopropyl β-D-1-thiogalactopyranoside (IPTG). The cultures were then incubated at 16 °C with shaking at 120 rpm for 16 h.

### 2.2. De Novo Genome Sequencing and Assembly

Genomic DNA was extracted from eight *S. commune* monokaryotic strains (20R-7-F01-S4y, -S17, -S5y, -S192, -S103, -S180, -S206, and -S10), representing different mating types. The DNA was fragmented to generate sequencing libraries with an average insert size of approximately 500 bp. Next-generation sequencing (NGS) was performed on the Illumina NovaSeq PE150 platform at APRxBIO (Shanghai, China). Raw sequencing data were initially processed using FASTQC software (version 0.12.1) to remove adapter sequences and low-quality reads (Q < 20) [33]. Clean reads were then assembled into scaffold-level contigs using SPAdes (version 3.12.0) [34]. The draft genome was constructed after gap closure using Gapcloser (version 1.12). Assembly integrity was assessed with BUSCO software (version 3.0.2) [35], and the overall assembly quality was evaluated using QUAST software (version 5.3.0) [36].

### 2.3. Phylogenetic Analysis and Annotation of Mating-Type Genes

Gene functions of the 20R-7-F01 monokaryotic strains were annotated using several databases, including the NCBI non-redundant protein database (Nr) (ftp://ftp.ncbi.nlm.nih.gov/blast/db/FASTA/nr.gz, accessed on 9 September 2022), Eukaryotic Clusters of Orthologous Groups (KOG, ftp://ftp.ncbi.nih.gov/pub/COG/KOG/kyva, accessed on 9 September 2022), SwissProt (ftp://ftp.uniprot.org/pub/databases/uniprot/current_release/knowledgebase/complete/uniprot_sprot.fasta.gz, accessed on 9 September 2022), Gene Ontology (GO, http://geneontology.org/docs/download-ontology/, accessed on 9 September 2022), Kyoto Encyclopedia of Genes and Genomes (KEGG, https://www.kegg.jp/kegg/download/, accessed on 9 September 2022), TIGR defined protein families (TIGRFAM) (version 15.0), and protein families (PFAM) database (version 35.0) [37]. The mating-type genes of each strain were initially annotated by conducting BLASTP (version 2.2.26) homology searches against the GeneBank non-redundant databases [38]. Multiple sequence alignment of the HD gene sequences was performed using ClustalX (version 2.1), with the non-conserved N-terminal regions excluded prior to analysis [39]. Phylogenetic analysis was performed using the Neighbor-Joining method in MEGA (version 6.0), and a phylogenetic tree was constructed to characterize the HD genes of the monokaryotic strains [40]. The nomenclature and annotation of the HD genes were subsequently assigned based on the phylogenetic results.

### 2.4. Protein–Protein Interaction (PPI) Prediction Analysis

The four HD proteins (R1, S4, V1, and Q4) analyzed in this study were modeled using the high-precision protein structure prediction tool AlphaFold (version 2.0.1) [41]. Potential protein interactions at the *S. commune* Aβ sublocus were predicted using the AI-PPI deep learning model developed by Pronetbio (Nanjing, China), which assesses interaction probabilities. The AI-PPI model integrates data from multiple public repositories, including the BioGRID database (version 4.4), STRING database (http://cn.string-db.org, accessed on 16 June 2023), UniProt (http://www.uniprot.org, accessed on 16 June 2023), KEGG (https://www.kegg.jp/kegg/download/, accessed on 16 June 2023), NCBI (https://www.ncbi.nlm.nih.gov/, accessed on 16 June 2023), and Gene Ontology (http://geneontology.org/docs/download-ontology/, accessed on 16 June 2023), to construct a comprehensive protein interaction network [42]. Interacting protein complexes were further analyzed with PDBePISA (https://www.ebi.ac.uk/pdbe/pisa/, accessed on 16 June 2023) to identify binding sites [43]. The predicted protein structures and interaction models were visualized using PyMOL (PyMOL Molecular Graphics System, version 2.5.5).

### 2.5. Polymerase Chain Reaction (PCR) Amplification of Mating-Type Genes

Four HD gene fragments (*abs4*-PV480567, *abq4*-PV480569, *abr1*-PV480561, and *abv1*-PV480563) were amplified via PCR from the *S. commune* 20R-7-F01 genome. The N-terminal half of EYFP (VN173, 1–519 bp) and the C-terminal half of EYFP (VC155, 520–772 bp) were PCR amplified from the EYFP plasmid [44,45,46]. PCR reactions were performed using an Applied Biosystems 2720 thermocycler (Thermo Fisher, Waltham, MA, USA), with Vazyme’s high-fidelity polymerase for all PCR-based cloning procedures. A list of the generated plasmids is provided in Appendix A, and the primers used are detailed in Appendix A.

### 2.6. Construction of BiFC Plasmids

The BiFC plasmids were constructed from the fungal expression vector pIG1783, which carries the *A. nidulans gpd* promoter, the *trpC* terminator, and the *hph* gene for selection of transgenic strains [31]. The plasmids were constructed using the same method as Subotić et al. [44]. In the first step, multiple cloning sites (MCSs), compatible with the synthesized dsDNA fragments, were enzyme-digested with NcoI (NEB) and NotI (NEB), and the green fluorescent protein tag carried by the original plasmid was removed. In the second step, the His-VN173, His-R1-VN173, and His-V1-VN173 fragments for N-terminal labeling of target genes were created by ligating the His-tag, VN173; His-tag, *abr1*, VN173, and His-tag, *abv1*, VN173 insert fragments, using primers PIG+His-F and PIG+VN173-R, respectively. In the third step, the HA-VC155, HA-S4-VC155 and HA-Q4-VC155 constructs for C-terminal labeling were generated by synthesizing the corresponding dsDNA fragments, which were then ligated into the vector backbone using primers PIG+HA-F and PIG+VC155-R, respectively (Appendix A). Finally, the fluorophore sequences were inserted into new vector backbones to yield the final BiFC plasmids: pIG1783-VN173, pIG1783-VC155, pIG1783-R1-VN173, pIG1783-V1-VN173, pIG1783-S4-VC155, and pIG1783-Q4-VC155 (Appendix A). Among them, pIG1783-VN173 and pIG1783-VC155 are empty vectors lacking target protein coding sequences and were used as negative controls in the BiFC experiments. The remaining four plasmids contain HD protein-coding sequences and were used for functional analysis.

### 2.7. Construction of Protein Expression Plasmids

Protein expression plasmids were constructed using the *E. coli* expression vector pET-28a^(+)^. Four dsDNA insert fragments, His-R1-VN173, His-V1-VN173, HA-S4-VC155, and HA-Q4-VC155, were synthesized and cloned into the vector, which was previously digested by NcoI (NEB) and BlpI (NEB) (Appendix A). This resulted in the creation of protein expression plasmids for the four HD genes: pET-28a^(+)^-His-R1, pET-28a^(+)^-His-V1, pET-28a^(+)^-HA-S4, and pET-28a^(+)^-HA-Q4 (Appendix A).

### 2.8. Protoplast Preparation and Transformation of S. commune

Protoplast preparation and polyethylene glycol (PEG)-mediated transformation were performed as described by Van Peer et al. [47]. Briefly, mycelia were incubated with a digestion solution containing 20 mg/mL *Trichoderma harzianum* lyase (Sigma, St. Louis, MO, USA) to facilitate protoplast formation. Subsequently, the BiFC plasmid was introduced into the protoplasts via PEG-mediated transformation to generate *S. commune* 20R-7-F01-S4y and 20R-7-F01-S5y transformants. These transformants were then used for BiFC interaction assays and functional verification, as detailed in Appendix A.

### 2.9. Mating Experiments and In Vivo Functional Validation

Mating experiments were conducted to determine the interactions and mating fertility of the four HD genes, *abs4*, *abq4*, *abr1*, and *abv1*. The transformed strains were co-cultured on modified MM medium, with a 10 mm distance between pairs of mating strains. The co-cultures were incubated in the dark at 30 °C for 3 days, followed by exposure to light at room temperature for 4 days to observe the formation of fruiting bodies.

### 2.10. Confocal Microscopy and Quantitative Image Analysis

Fluorescence imaging was performed using an LSM980 confocal microscope (Zeiss, Jena, Germany) equipped with 20 × and 40 × Neofluor objectives. The system utilized semiconductor lasers with the following wavelengths: 405 nm (30 mW), 488 nm (25 mW), 543 nm (25 mW), 594 nm (8 mW), and 639 nm (5 mW). Four fluorescence detection channels and one DIC transmitted light detection channel were available. EYFP fluorescence was captured using a 514 nm laser diode. The acquired images were processed and analyzed using ZEISS ZEN 3.1 (blue edition) [48], and the fluorescence intensity of the images was measured using ImageJ. The size of the measurement area for each image was exactly the same, and the image with the highest fluorescence intensity data was selected for normalization to obtain the relative fluorescence intensity. Raw images were acquired with consistent acquisition settings, including laser power, detector sensitivity, and pixel dwell time, across all samples.

### 2.11. His Pull-Down Assays and Western Blot Analyses

His pull-down assays were performed using the instructions for the Pierce ^TM^ Pull-Down PolyHis Protein: Protein Interaction Kit (Thermo Fisher) [49]. His-R1-VN173 and His-V1-VN173 fusion proteins, expressed in *E. coli* BL21 (DE3), were used as His-bait proteins, while HA-S4-VC155 and HA-Q4-VC155 fusion proteins served as prey proteins. The supernatants from the His pull-down elution samples were collected and stored at −80 °C for subsequent Western blot analysis. The Input groups that were not treated with His pull-down were set up as positive controls for the experimental groups, while the samples treated only with HA-tagged proteins were set up as negative controls.

For Western blotting, proteins were separated through 6% sodium dodecyl sulphate-polyacrylamide gel electrophoresis (SDS-PAGE) and transferred to 0.45 μm polyvinylidene fluoride (PVDF) membranes via a wet-transfer system (Tanon, Shanghai, China). The membranes were blocked with 5% skimmed milk for 1 h at room temperature and then incubated overnight at 4 °C with primary antibodies. After washing with Tris-buffered saline containing 0.1% Tween-20 (TBST), the membranes were incubated with horseradish peroxidase (HRP)-conjugated secondary antibodies for 1 h at room temperature, washed again with TBST, and developed using ECL Western Blotting Substrate (Vazyme, Nanjing, China). Protein detection and analysis were performed using a UVP Auto Chemi Image system (Tanon). The primary antibodies used were Rabbit anti-HA tag polyclonal antibody, Rabbit anti-His tag polyclonal antibody, and Goat anti-Rabbit IgG-HRP.

### 2.12. Statistical Analysis

All experiments included at least three biological replicates. Data are presented as the mean ± standard error. Statistical analysis was performed using SPSS 28.0.1.1 and ImageJ (version 1.8.0.112). Analysis of variance (ANOVA) was used to compare multiple treatment groups, followed by Tukey’s test (for equal variance) or the least significant difference (LSD) test (for unequal variance or greater sensitivity) for pairwise comparisons. Statistical significance was set at *p ≤* 0.05.

## 3. Results

### 3.1. Identification of the A Mating-Type Locus in S. commune 20R-7-F01 Strains

To identify the A mating-type loci, we analyzed eight monokaryotic strains of 20R-7-F01 (20R-7-F01-S4y, -S17, -S5y, -S192, -S103, -S180, -S206, and -S10) representing various mating types (AxBx, AxBy, AyBx, and AyBy) using homologous matching methods and sequencing data. Our findings revealed that all Ax mating-type strains (AxBx and AxBy) exhibited an identical genetic structure at the A mating-type locus (Figure 1a). This locus consists of two subloci, Aα and Aβ, each encoding distinct HD1 and HD2 proteins. The Aα sublocus contains the *aaz4* and *aay4* genes, which encode the Z4 (HD1) and Y4 (HD2) proteins, respectively. The Aβ sublocus includes four additional genes encoding HD1 (S and Q) and HD2 (R and V) proteins, organized into two “head-to-head” pairs (Figure 1a, Appendix A). These genes are homologous to the *abr6*, *abs6*, *abv6*, and *abq6* genes described in the H4-8 strain (Appendix A), but the similarity is not high [2]. Therefore, given the unresolved specificity of the Aβ sublocus genes in strain 20R-7-F01 mating-compatible strains, we applied a provisional Aβ4-specific numbering system to differentiate these four genes, naming them *abr4*, *abs4*, *abv4*, and *abq4*.

Similarly, all Ay mating-type strains (i.e., AyBx and AyBy) in strain 20R-7-F01 exhibited identical structures at the A mating-type locus (Figure 1b). These loci share a similar overall organization; however, the Aα sublocus in the Ay mating type encodes a single Y protein, represented by the gene *aay1*. Although the Aβ subloci in both Ay and Ax mating types share conserved gene orientation and function, they require distinct specificity numbering. To differentiate the Ay mating-type genes, we applied Aβ1-specific numbering, designating these genes as *abr1*, *abs1*, *abv1*, and *abq1* (Figure 1b, Appendix A).

Phylogenetic analysis of the amino acid sequences encoding HD1 and HD2 proteins revealed significant genetic divergence between the two protein groups, with each forming distinct evolutionary branches (Figure 1c, Appendix A). Within the HD1 branch, the S, Q, and Z proteins clustered into separate evolutionary groups, with S and Q proteins showing a closer relationship. Similarly, the R, V, and Y proteins formed independent evolutionary branches within the HD2 group, with R and V proteins being more closely related. These findings were consistent with the classification of the subloci. In other words, HD genes at the same sublocus are more similar.

This comprehensive genetic analysis confirms the complexity of the A mating-type locus in *S. commune*, and the application of distinct numbering systems for the Aβ subloci further clarifies the functional and evolutionary relationships among these genes.

### 3.2. PPI Prediction of Aβ Mating-Type HD Proteins

Using the principle of non-self-recognition between HD1 and HD2 proteins at the A mating-type locus, we predicted several potential PPIs within the Aβ subloci of *S. commune* 20R-7-F01. Specifically, based on the premise that same-specific HD proteins cannot be recognized, we chose four pairs, R1-S4, R1-Q4, V1-S4, and V1-Q4, for validation, as illustrated by the green dashed lines in Figure 2a.

Structural analysis of these interactions revealed high confidence levels for the pairs: 95.21% for R1-S4, 97.14% for R1-Q4, 95.93% for V1-S4, and 97.43% for V1-Q4 (Figure 2b–e). Based on these predictions, we propose that the four HD proteins within the Aβ sublocus likely form four distinct protein complexes. Further examination of the interaction sites identified high-confidence binding regions, as illustrated in Appendix A. These findings suggest that the HD proteins within the Aβ sublocus play a pivotal role in regulating sexual reproduction at the A mating-type locus through their involvement in protein–protein interactions.

### 3.3. BiFC Interaction of Aβ Mating-Type HD Proteins

To validate the predicted protein interactions, we employed the in vivo BiFC system in *S. commune* 20R-7-F01, as outlined in Figure 3a. Strains expressing only the VN173 or VC155 empty vectors did not exhibit yellow fluorescence in the monokaryotic state, nor did they produce fluorescence when mated with each other (Appendix A). After addition of the target proteins, fluorescence microscopy revealed that none of the individual strains expressing R1-VN173, V1-VN173, S4-VC155, and Q4-VC155 displayed yellow fluorescence (Figure 3b–e), confirming the absence of autofluorescence or fluorescence from single expression plasmids. Upon mating, strains expressing R1-VN173 and S4-VC155 emitted yellow fluorescence (Figure 3b and Appendix A), indicating a successful interaction. A similar result was observed for the mating of strains expressing V1-VN173 and Q4-VC155, which also produced yellow fluorescence (Figure 3e and Appendix A). These observations suggest that the R1-S4 and V1-Q4 protein pairs interact in vivo in *S. commune* 20R-7-F01. In contrast, no fluorescence was detected in the hyphae of the mating combinations R1-VN173 × Q4-VC155 (Figure 3c and Appendix A) and V1-VN173 × S4-VC155 (Figure 3d and Appendix A), indicating that these protein pairs do not interact under the tested conditions.

### 3.4. His Pull-Down Assays and Western Blot Analysis of Aβ Mating-Type HD Proteins

To further validate the interaction between the protein pairs R1-S4 and V1-Q4, we performed in vitro His pull-down assays and observed protein interactions using Western blot analysis. Western blot analysis revealed that R1-His migrated as a single band at approximately 130 kDa, S4-HA at approximately 100 kDa, V1-His at approximately 110 kDa, and Q4-HA at approximately 90 kDa, confirming the successful expression of all four proteins in vitro. The results from the Input groups confirmed that, in the absence of His pull-down, R1-His and V1-His were successfully expressed and detected using the His antibody assay, while both S4-HA and Q4-HA were clearly expressed in the HA antibody assay. Furthermore, the HA negative controls showed no non-specific bands, indicating the specificity of the interaction assays (Figure 4). In the His pull-down assay, the R1-His bait protein specifically captured the S4-HA protein (Figure 4a), whereas the V1-His bait protein did not interact with S4-HA but specifically captured the Q4-HA protein (Figure 4b). These findings are in agreement with the results from the in vivo BiFC assays, further supporting the interaction between the R1-S4 and V1-Q4 protein pairs.

### 3.5. Validation of In Vivo Interactions for Fertility

It is well established that *S. commune* 20R-7-F01 monokaryotic strains can only produce fruiting bodies when the A and B mating-type factors are distinct (i.e., A ≠ B ≠) (Figure 5a,e), and that A = B ≠ mating combinations result in a “flat” interaction phenotype (Figure 5b,f). To validate the in vivo interactions of the R, S, V, and Q proteins, we utilized two sets of 20R-7-F01 strains (AyBy × AyBx and AxBx × AxBy), which exhibit A = B ≠ mating combinations, as the experimental system. The AyBy and AyBx strains of *S. commune* 20R-7-F01 carry genes encoding the R1, S1, V1, and Q1 proteins. BiFC plasmids containing the S4 and Q4 genes (pIG1783-S4-VC155 and pIG1783-Q4-VC155) were transfected into strain 20R-7-F01-S5y (AyBy). Following transformation, mating assays were performed with the wild-type 20R-7-F01-S206 (AyBx) strain (Figure 5c,d). The mating results showed the production of primordia, which was completely different from the mating results of the wild-type A = B ≠ mating combination that failed to accomplish sexual compatibility and development (Figure 5b).

Similarly, the AxBx and AxBy strains of *S. commune* 20R-7-F01 contain genes encoding the R4, S4, V4, and Q4 proteins. BiFC plasmids carrying the R1 and V1 genes (pIG1783-R1-VN173 and pIG1783-V1-VN173) were transfected into the 20R-7-F01-S4y (AxBx) strain, and mating assays were conducted with the wild-type 20R-7-F01-S103 (AxBy) strain (Figure 5g,h). Again, the wild-type non-fertile phenotype (Figure 5f) was converted into a fertile phenotype.

In both mating experiments, the transformed strains displayed the characteristic “flat” colony morphology associated with A = B ≠ mating combinations, with a lack of aerial mycelia, mycelia growing adherent to the surface of the medium, and a clear demarcation line between the two mycelial sides. However, these strains exhibited the appearance of *S. commune* fruiting bodies, indicating successful fertility restoration (Figure 5). The same results were obtained through microscopic examination, with pseudo clamp connections appearing in the transformed monokaryotic strains and a few clamp connections being formed in the transformed mating strains (Appendix A). These results provide direct evidence that the R, S, V, and Q proteins interact and are functionally involved in sexual compatibility and fruiting body development in *S. commune*. Moreover, they demonstrate that R-S and V-Q protein pairs can form non-self-recognizing protein complexes during sexual reproduction, playing a critical role in the activation of the sexual compatibility and fruiting body development pathway at the A mating-type locus.

## 4. Discussion

Our findings provide new insights into the structure and functional mechanism of the A mating-type locus in *S. commune*. At the Aβ sublocus, we identified four distinct HD proteins: R, S, V, and Q. These HD proteins share homologous structural domains known to be involved in sequence-specific DNA binding, a characteristic feature of proteins that regulate eukaryotic developmental genes [22]. In strain 20R-7-F01, the four HD genes at the Aβ sublocus are arranged as two HD1-HD2 “head-to-head” pairs, consistent with the arrangement of Y-Z proteins in the Aα sublocus [2].

Protein structure prediction analysis revealed that the R (HD2), S (HD1), V (HD2), and Q (HD1) proteins could potentially form four distinct protein interaction pairs: i.e., R-S, R-Q, V-S, and V-Q. However, both in vivo and in vitro studies demonstrate that only the R-S and V-Q protein pairs, organized in a “head-to-head” configuration, exhibit non-self-combinatorial interactions and are capable of activating the developmental pathway regulated by the A mating-type locus.

A particularly intriguing observation arises from our inability to detect functional interactions between the R-Q and V-S protein pairs, despite their proximity at the Aβ sublocus and minimal gene spacing. Notably, the R and V proteins, as well as the S and Q proteins, show significant similarities in both gene structure and domain homology. Protein structure models generated using AlphaFold indicate over 95% confidence in the predicted interactions for all four potential protein pairs. Consistent with previous work by Yue et al., who noted that the Y protein recognition site spans residues 10–80, we compared the amino acid sequences of the R, S, V, and Q proteins within the 1–130 region of strain 20R-7-F01 (Appendix A) [50]. Our comparison revealed that the R4 and R1 proteins share identical or highly similar amino acid sequences in this region, as do the other three HD proteins. However, the R-V and S-Q protein pairs displayed marked differences in their sequences. This suggests that the mutual recognition among the *S. commune* HD1-HD2 proteins may follow a more intricate pattern, potentially involving specific recognition sites within the N-terminal short sequences or relying on the structural conformation of the folded proteins. We hypothesize that while the R and V proteins, as well as the S and Q proteins, may form structurally similar binding regions, they fail to interact during recognition. If this is the case, mutations in conserved sequences within the recognition regions of the HD genes could lead to the emergence of new mutually recognizable HD gene pairs, thereby increasing the diversity of the *S. commune* mating-type system.

In addition, dimerization and heterologous binding of HD proteins are mainly driven by the arrangement of hydrophobic and polar residues on the conserved α-helical interface, which form highly specific interface topologies through van der Waals forces and hydrogen bonds (such as residues 47 and 51 in the Oct-1 POU structure docking DNA and mediating protein–protein interactions) [51]. Side chain charges and local pKa shifts (such as upward pKa shifts of lysine and arginine on the binding surface) enhance interfacial electrostatic attraction, thereby stabilizing the interaction between HD domains, a mechanism that is also conserved in multiple HD family members through pKa perturbations [52]. Binding of specific partner proteins (such as Extradenticle and Homothorax) may induce conformational rearrangement of the HD domain, exposing or reshaping key interaction interfaces and significantly improving the binding affinity and specificity of the HD protein [53].

Stamberg and Koltin proposed the existence of 32 Aβ sublocus specificities in nature; however, the precise number of Aβ subloci within the mating-compatible strains of *S. commune* remains unclear [54]. In this study, we tentatively assigned Aβ4 and Aβ1 to these subloci for ease of differentiation. Comparative analysis of the Aβ sublocus gene structures in the current study and those resolved by Ohm et al. reveals that *S. commune* 20R-7-F01 lacks the T and U proteins, complicating protein nomenclature [2]. Notably, the Q protein identified in this study shares significant homology with both T and U, which influenced our decision to designate it as Q. This naming choice was further supported by phylogenetic analysis, although it is possible that the position and characteristics differ from those previously described for Q proteins by Ohm et al. [2].

Despite these uncertainties, our findings firmly establish the roles of R and S, along with V and Q, in regulating sexual compatibility and fruiting body development in *S. commune*. Unlike the hypotheses proposed by Shen et al. and others [25], we directly validated the involvement of these proteins in sexual compatibility through exogenous gene transformation. However, unlike Luo et al., we were unable to determine which HD proteins at the Aβ sublocus contain nonessential homology domain sequences or identify which HD protein plays a more critical role at this sublocus [22]. Future studies are needed to explore these questions in greater depth, providing a clearer understanding of the specific contributions of these HD protein families to mating compatibility.

## 5. Conclusions

This study investigates the Aβ mating-type sublocus in *S*. *commune* and its role in sexual compatibility. Four HD genes were identified, encoding two HD1 proteins (S and Q) and two HD2 proteins (R and V), arranged head-to-head. Functional analyses demonstrated that interactions between the R-S and V-Q protein pairs are critical for activating the sexual reproduction pathway and promoting fruiting body formation. These findings reveal that, similar to the Aα sublocus, the Aβ sublocus is essential for coordinating HD protein activity during fungal mating. This research not only advances our understanding of the molecular basis of fungal mating systems but also provides valuable insights into population diversity and breeding strategies for edible fungi.

## Figures and Tables

**Figure 1 jof-11-00451-f001:**
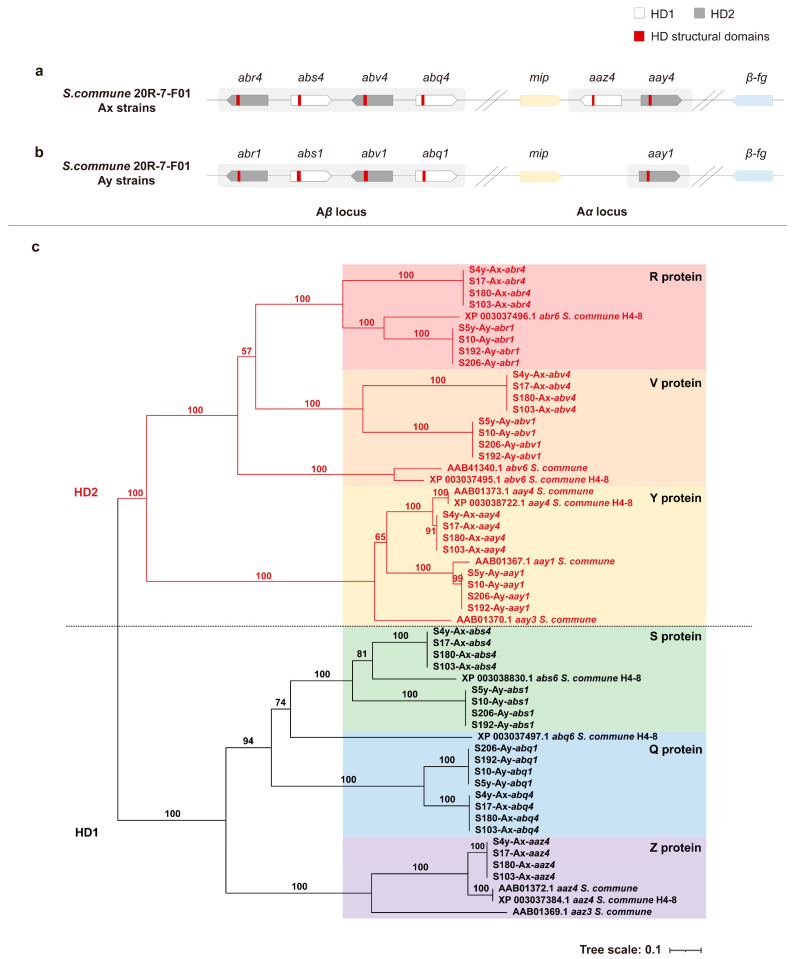
Structure and phylogeny of the A mating-type locus in *S. commune* 20R-7-F01 strains. Genes encoding HD1 and HD2 proteins in (**a**) Ax mating type in 20R-7-F01; (**b**) Ay mating type in 20R-7-F01; (**c**) phylogenetic tree of HD genes. Gene structure diagrams illustrate HD1 proteins (white boxes), HD2 proteins (grey boxes), and HD structural domains (red boxes). In the phylogenetic tree (**c**), the red branch represents the HD2 lineage, while the black branch represents the HD1 lineage.

**Figure 2 jof-11-00451-f002:**
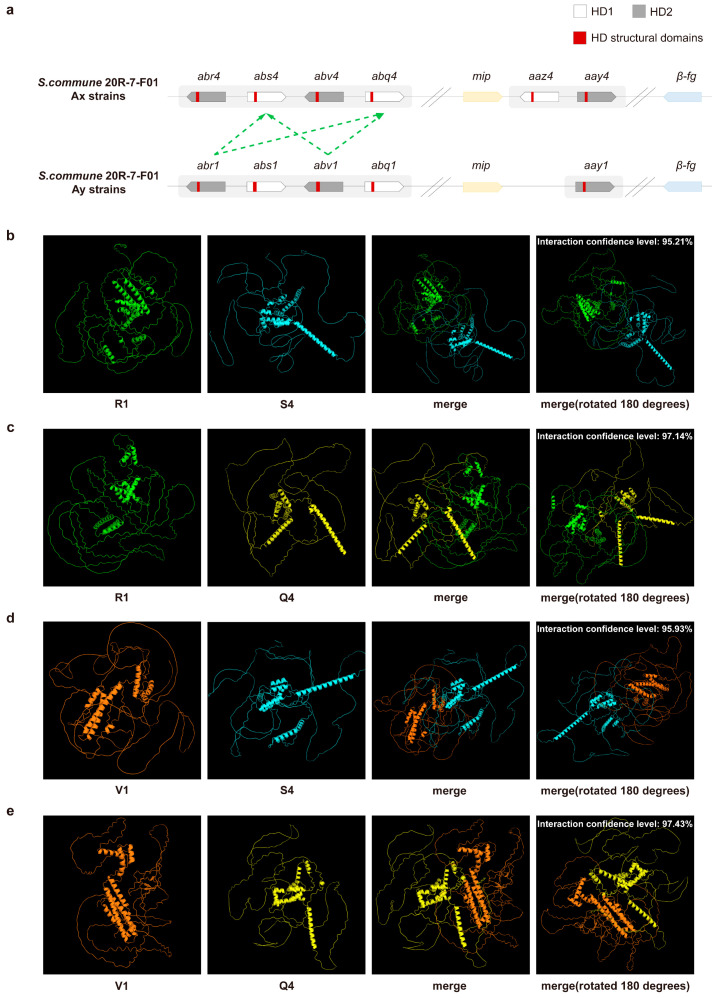
Analysis of the HD interaction proteins at the Aβ sublocus. (**a**) Putative HD protein interaction combinations at the Aβ sublocus. Structures of (**b**) R1 (*abr1*, green) and S4 (*abs4*, blue); (**c**) R1 (*abr1*, green) and Q4 (*abq4*, yellow); (**d**) V1 (*abv1*, orange) and S4 (*abs4*, blue); (**e**) V1 (*abv1*, orange) and Q4 (*abq4*, yellow) proteins. Structures were visualized and aligned using the PyMOL Molecular Graphics System (version 2.5.5). The green dashed arrows indicate HD protein pairs that may interact with each other.

**Figure 3 jof-11-00451-f003:**
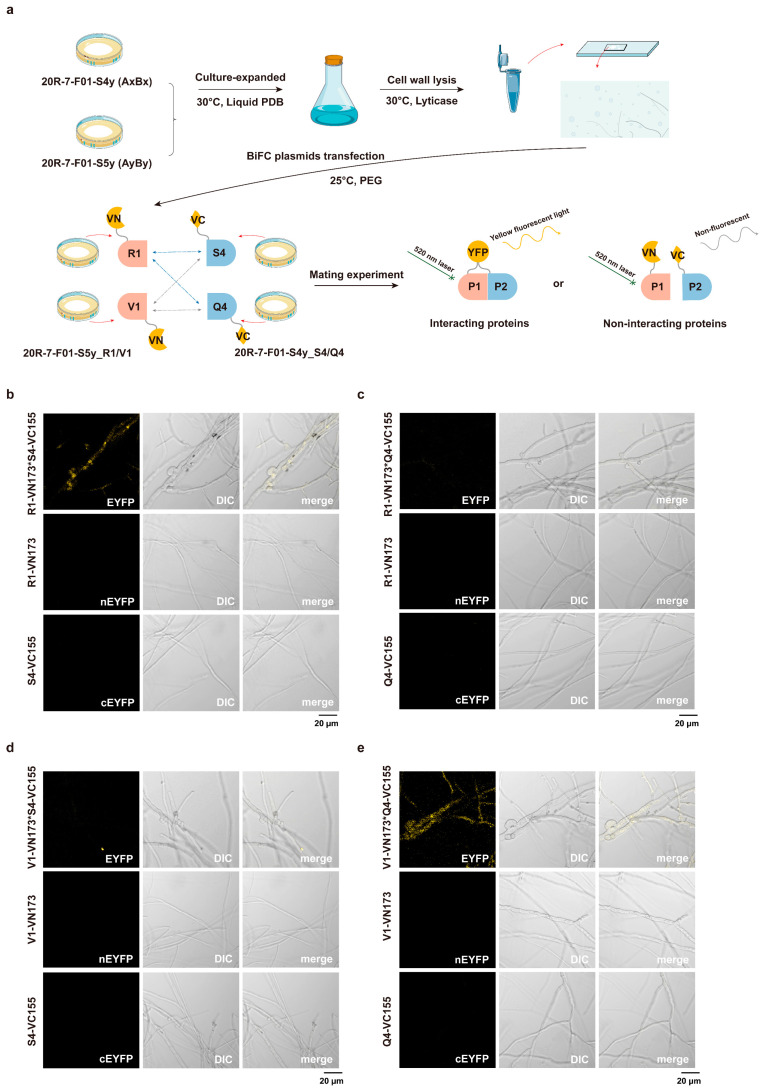
BiFC system visualization of HD protein interactions. (**a**) Visualization workflow for BiFC analysis of HD protein interactions in *S. commune* 20R-7-F01. BiFC fluorescence images showing in vivo interaction between (**b**) R1-VN173 and S4-VC155; (**c**) R1-VN173 and Q4-VC155; (**d**) V1-VN173 and S4-VC155; and (**e**) V1-VN173 and Q4-VC155. Bar = 20 μm.

**Figure 4 jof-11-00451-f004:**
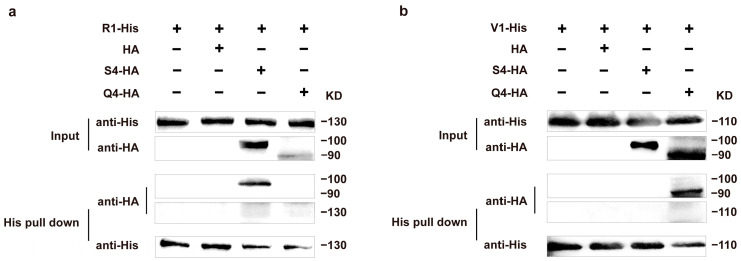
His pull-down assay and immunoblot analysis for in vitro HD protein interactions. Proteins expressed in *E. coli* were analyzed for interaction using immunoblotting with anti-His and anti-HA antibodies. Pull-down assay of (**a**) R1 protein with S4, Q4 proteins and (**b**) V1 protein with S4, Q4 proteins.

**Figure 5 jof-11-00451-f005:**
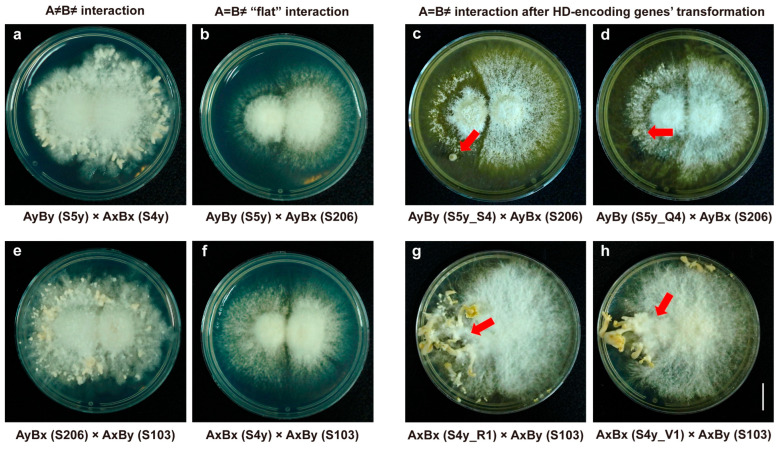
In vivo functional validation of HD interacting proteins in *S. commune*. Mating between (**a**) wild-type AyBy (S5y) and AxBx (S4y) strains; (**b**) wild-type AyBy (S5y) and AyBx (S206) strains; (**c**) AyBy (S5y) strain transfected with exogenous *abs4* gene and wild-type AyBx (S206) strain; (**d**) AyBy (S5y) strain transfected with exogenous *abq4* gene and wild-type AyBx (S206) strain; (**e**) wild-type AyBx (S206) and AxBy (S103) strains; (**f**) wild-type AxBx (S4y) and AxBy (S103) strains; (**g**) AxBx (S4y) strain transfected with exogenous *abr1* gene and wild-type AxBy (S103) strain; and (**h**) AxBx (S4y) strain transfected with exogenous *abv1* gene and wild-type AxBy (S103) strain. The red arrow indicates the primordia and fruiting bodies. Bar = 1 cm.

## Data Availability

The original contributions presented in this study are included in the article/Appendix A; further inquiries can be directed to the corresponding authors. The accession numbers of the eight *S. commune* samples are 20R-7-F01-S4y (JBMBGP000000000), 20R-7-F01-S5y (JBMBGQ000000000), 20R-7-F01-S103 (JBMBGR000000000), 20R-7-F01-S206 (JBMBGS000000000), 20R-7-F01-S17 (JBMBGT000000000), 20R-7-F01-S192 (JBMBGU000000000), 20R-7-F01-S180 (JBMBGV000000000), and 20R-7-F01-S10 (JBMBGW000000000).

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
