# Peer review of "Characterization of Homeodomain Proteins at the Aβ Sublocus in Schizophyllum commune and Their Role in Sexual Compatibility and Development"

_jof, 2025, doi:10.3390/jof11060451_

Round 1
Reviewer 1 Report
Dear Authors,
I have reviewed the manuscript entitled "Characterization of Homeodomain Proteins at the Aβ Sublocus in Schizophyllum commune and Their Role in Sexual Compatibility and Development" and found it well-written and of interest for the scientific community as it underlines the important role of the Aß sublocus in fungal reproduction and furnishes valuable findings for breeding edible and medicinal S. commune strains.
Because the data presented in the manuscript is generally well-organized and the manuscript is well-written there are only a few very small observations to be taken into consideration to improve the quality of the manuscript:
- Figures 1-3 are a bit too small and difficult to read, therefore they could be slightly enlarged to ensure a better visualization and readability.
- Please make sure that each abbreviation is detailed the first time it appears in the manuscript (e.g. Line 109: Integrated ocean drilling program (IODP)).
- Line 32: I suggest using " These combinations" instead of "This cobination" .
- Line 342: Please correct "Maing" to "Mating".
Author Response
Reviewer 1
Comments 1:
Figures 1-3 are a bit too small and difficult to read, therefore they could be slightly enlarged to ensure a better visualization and readability.
Response 1: We agree with this comment. We have adjusted the font size in Figures 1-3 and adjusted the typography of the images to ensure better visibility and readability.
Figure 1 Line 291
Figure 2 Line 312
Figure 3 Line 334
Comments 2:
Please make sure that each abbreviation is detailed the first time it appears in the manuscript (e.g. Line 109: Integrated ocean drilling program (IODP)).
Response 2: We agree with this comment and have reviewed and revised the entire article. The specific revisions are:
Line 119-120: the International Ocean Discovery Program (IODP)
Line 129-130: isopropyl β-D-1-thiogalactopyranoside (IPTG)
Line 147-149: Kyoto Encyclopedia of Genes and Genomes (KEGG), TIGR defined protein families (TIGRFAM), and protein families (PFAM) database
Line 204: polyethylene glycol (PEG)
Line 239-240: sodium dodecyl sulphate-polyacrylamide gel electrophoresis (SDS-PAGE)
Line 253: Analysis of variance (ANOVA)
Line 254: least significant difference (LSD)
Line 498: The above abbreviations have also been added at Abbreviations.
Comments 3:
Line 32: I suggest using " These combinations" instead of "This cobination".
Response 3: We agree with this comment and have made the following correction (the original line number has changed due to the addition of some content to the introduction):
Line 32: This combination → Line 45: These combinations
Comments 4:
Line 342: Please correct "Maing" to "Mating".
Response4: We agree with this comment and have made the following correction (the original line number has changed due to the addition of some content to the introduction):
Line 342: Maing → Line 380: Mating
Reviewer 2 Report
The study explores the function of homeodomain (HD) proteins encoded by the Aβ sublocus in the fungus Schizophyllum commune, focusing on their roles in sexual compatibility and fruiting body development. Using a robust methodological combination, including in silico protein modeling, in vitro interaction assays, and in vivo functional validation, the authors identified four HD-encoding genes (HD1 and HD2 types), organized in head-to-head pairs. The findings demonstrate that the R-S and V-Q pairs interact functionally to regulate mating compatibility.
This work contributes significantly to our understanding of the genetic and molecular mechanisms underlying fungal reproduction and provides a promising basis for future breeding strategies in edible and medicinal mushrooms.
However, several key aspects require revision to meet the standards of reproducibility, clarity, and analytical rigor. Although the data presented support the main conclusions, substantial revisions are needed, especially in the methods and quantitative analysis of the results, to meet the reproducibility criteria and strengthen the discussion.
Throughout the manuscript, Latin terms such as in vivo and in vitro should be consistently italicized, in accordance with scientific writing conventions.
I recommend that the authors provide additional quantitative data regarding the interaction and fertility assays, include appropriate functional controls, and offer a more in-depth discussion of the molecular determinants driving HD protein interactions.
Introduction
Mainly, the authors should make a clearer effort to contextualize why it is important to study the sexual reproduction system in Schizophyllum commune. While the manuscript focuses on genetic mechanisms and compatibility regulation, it lacks a discussion of the broader biological or economic relevance of this species. Is S. commune considered a model organism for fungal development? Does it pose any pathogenic risk to crops, humans, or stored materials? Does it hold industrial potential as an edible or medicinal mushroom that justifies the development of compatible or improved strains?
The introduction is generally well-written but would benefit from:
- A clearer emphasis on the scientific gap being addressed.
- Avoiding early disclosure of results.
- A stronger articulation of the functional relevance of the Aβ sublocus in fungal biotechnology and genetics.
The last sentence would be a thought-provoking call focusing on the scientific gap. Suggested final sentence:
“In this study, we aim to address a critical gap in the understanding of the molecular mechanisms underlying sexual reproduction in S. commune, focusing on the poorly characterized Aβ sublocus, which may be essential for genetic improvement strategies.”
Materials and Methods
The selection criteria for the eight derived isolates are unclear.
BiFC and pull-down assays lack negative and positive control details.
The number of biological and technical replicates is not specified.
Statistical tests used for comparisons are not described.
Autofluorescence controls using empty vectors were not mentioned.
Lines 114–117: The description of E. coli expression strains is appropriate, but lacks details such as induction OD600 and expression time.
Lines 200–207: While microscopy settings are described, the method for fluorescence quantification and normalization (e.g., by area or intensity threshold) should be explained.
Results
The figures are extremely difficult to interpret due to their small size, low resolution, and the use of excessively small font labels, which significantly hampers readability and data comprehension.
BiFC images are presented without quantification of signal intensity (e.g., mean fluorescence in regions of interest with standard error).
The pull-down assays and Western blot analyses lack proper negative controls (e.g., a non-interacting protein or untagged version), which would help confirm specificity.
Author Response
Reviewer 2
Comments 1:
Throughout the manuscript, Latin terms such as in vivo and in vitro should be consistently italicized, in accordance with scientific writing conventions.
Response 1:
We agree with your comments. We have revised in vivo and in vitro to be uniformly italicized throughout the manuscript, with the line numbers of the revision listed as:
in vivo: Line 16, 110, 319, 330, 336-337, 352, 363, 380, 410.
in vitro: Line 341, 345, 355, 410.
Comments 2:
I recommend that the authors provide additional quantitative data regarding the interaction and fertility assays, include appropriate functional controls, and offer a more in-depth discussion of the molecular determinants driving HD protein interactions.
Response 2: We agree with this review and have added the following to the Discussion section (Line 433-444): “In addition, dimerization and heterologous binding of HD proteins are mainly driven by the arrangement of hydrophobic and polar residues on the conserved α-helical interface, which form highly specific interface topologies through van der Waals forces and hydrogen bonds (such as residues 47 and 51 in the Oct-1 POU structure docking DNA and mediating protein-protein interactions) [51]. Side chain charges and local pKa shifts (such as upward pKa shifts of lysine and arginine on the binding surface) enhance interfacial electrostatic attraction, thereby stabilizing the interaction between HD domains, a mechanism that is also conserved in multiple HD family members through pKa perturbations [52]. Binding of specific partner proteins (such as Extradenticle and Homothorax) may induce conformational rearrangement of the HD domain, exposing or reshaping key interaction interfaces and significantly improving the binding affinity and specificity of the HD protein [53].” The additional content provides in-depth insights into the molecular determinants that regulate HD protein interactions, which greatly enhances the rigor and clarity of our study.
Comments 3:
Mainly, the authors should make a clearer effort to contextualize why it is important to study the sexual reproduction system in Schizophyllum commune. While the manuscript focuses on genetic mechanisms and compatibility regulation, it lacks a discussion of the broader biological or economic relevance of this species. Is S. commune considered a model organism for fungal development? Does it pose any pathogenic risk to crops, humans, or stored materials? Does it hold industrial potential as an edible or medicinal mushroom that justifies the development of compatible or improved strains?
Response 3: The first part of the introduction does not describe Schizophyllum commune in enough detail, thank you for pointing this out. We agree with this comment. Therefore, we re-describe this part, replacing the original “The tetrapolar fungus Schizophyllum commune is a rare edible and medicinal mushroom, with high nutritional and medicinal values [1-3]” with “The tetrapolar fungus Schizophyllum commune is a typical white-rot fungus belonging to the genus Schizophyllum, family Schizophyllaceae, order Agaricales, class Agaricomycetes, and phylum Basidiomycota [1]. As a model organism of the genus Schizophyllum, it has been extensively studied in fungal genetics and developmental biology [2]. Additionally, S. commune is a rare edible and medicinal mushroom with high nutritional and pharmacological value [3-5]. It produces various valuable metabolites, including schizophyllan, which exhibits immunomodulation, antitumor activity, and wound-healing properties [6]. The bioactive compounds possess antioxidant and anti-inflammatory activities by neutralizing free radicals and reducing oxidative stress [7]. Its fermentation products are widely used as additives in the food and feed industries, underscoring its substantial industrial value [8]. However, if used improperly, S. commune may lead to pathogenic outcomes: it can infect fruit trees and cause plant diseases[9], and inhalation of its spores may cause respiratory infections in humans[10]. Therefore, developing compatible or improved strains of S. commune is essential for safe and beneficial applications.” This section is now located on Line 25-38.
Comments 4:
The introduction is generally well-written but would benefit from:
A clearer emphasis on the scientific gap being addressed.
Avoiding early disclosure of results.
A stronger articulation of the functional relevance of the Aβ sublocus in fungal biotechnology and genetics.
Response 4: We agree with this comment. Therefore, we re-describe this part, replacing the original “In this study, we performed a comprehensive analysis of the A mating-type locus in eight monokaryotic S. commune strains derived from the parental dikaryotic strain 20R-7-Z01. Through gene sequencing, protein structure prediction, and protein interactions validation techniques, we identified two novel pairs of HD proteins at the Aβ sublocus: R (HD2)-S (HD1) and V (HD2)-Q (HD1), are capable of forming non-self-combination interactions. These interactions are reminiscent of the Y-Z recognition pattern observed at the Aα sublocus. Additionally, through exogenous transformation of monokaryotic strains, two wild-type strains exhibiting A=B≠ mating incompatibility were transformed into mating-compatible and capable of fruiting bodies formation. This provides direct evidence that the Aβ sublocus is involved in regulating sexual compatibility in S. commune. These findings significantly enhance our understanding of the genetic structure and physiological functions of the Aβ sublocus, and offer new insights into the diversity and breeding of edible S. commune strains.” with “Investigating the Aβ sublocus offers a more refined model for elucidating the mechanisms of sexual reproduction in fungi and provides valuable insights for breeding and functional gene discovery in edible and medicinal mushrooms. Once the genetic structure of the Aβ sublocus is characterized, molecular markers can be employed to screen for desirable traits, and targeted modifications may be achieved through gene editing technologies. Therefore, in this study, we aim to address a critical gap in the understanding of the molecular mechanisms underlying sexual reproduction in S. commune, focusing on the poorly characterized Aβ sublocus, which may be essential for genetic improvement strategies.” Now the location of this sentence is Line 99-107.
Comments 5:
The last sentence would be a thought-provoking call focusing on the scientific gap. Suggested final sentence:
“In this study, we aim to address a critical gap in the understanding of the molecular mechanisms underlying sexual reproduction in S. commune, focusing on the poorly characterized Aβ sublocus, which may be essential for genetic improvement strategies.”
Response 5: We agree with this comment. We changed the last sentence of Introduction from “These findings significantly enhance our understanding of the genetic structure and physiological functions of the Aβ sublocus, and offer new insights into the diversity and breeding of edible S. commune strains.” to “Therefore, in this study, we aim to address a critical gap in the understanding of the molecular mechanisms underlying sexual reproduction in S. commune, focusing on the poorly characterized Aβ sublocus, which may be essential for genetic improvement strategies.” (Line 104-107)
Comments 6:
The selection criteria for the eight derived isolates are unclear.
Response 6: We agree with this comment. Here, we did not clearly describe the selection criteria for the eight derived isolates. Therefore, we have added selection criteria for these eight strains to Materials and Methods: These eight monokaryotic strains constituted a complete set of mating subtype strains, representing the full range of monokaryotic strains isolated from strain 20R-7-ZF01[26]. (Line 117-118)
Comments 7:
BiFC and pull-down assays lack negative and positive control details.
Response 7: We agree with this comment and have added a detailed description of the BiFC empty vectors, as well as the negative and positive controls used in the His pull-down experiments, into the Materials and Methods section. Additionally, we have added the corresponding fluorescence results of the BiFC negative control and the control results of the His pull-down assays to the Results section. Supporting data for the BiFC negative controls have also been provided in the Supplementary Materials:
BiFC assay, Material and Methods: “Among these, pIG1783-VN173 and pIG1783-VC155 are empty vectors lacking target protein coding sequences and were used as negative controls in the BiFC experiments. The remaining four plasmids contain HD protein coding sequences and were used for functional analysis.” (Line 192-195) Results: “Strains expressing only the VN173 or VC155 empty vectors did not exhibit yellow fluorescence in the monokaryotic state, nor did they produce fluorescence when mated with each other (Figure S3).” (Line 320-322) Supplementary Material: “Figure S3-add BiFC fluorescence images showing in vivo interaction between VN173 and VC155 empty vectors.”
Pull-down assay, Materials and Methods: “The Input groups that were not treated with His pull-down were set up as positive controls for the experimental groups, while the samples treated only with HA-tagged proteins were set up as negative controls.” (Line 236-238) Results: “The results from the Input groups confirmed that, in the absence of His pull-down, R1-His and V1-His were successfully expressed and detected using the His antibody assay, while both S4-HA and Q4-HA were clearly expressed in the HA antibody assay. Furthermore, the HA negative controls showed no non-specific bands, indicating the specificity of the interaction assays (Figure 4).” (Line 345-349)
Comments 8:
The number of biological and technical replicates is not specified.
Response 8: We agree with this comment and have supplemented the number of biological and technical replicates involved in this experiment in the Materials and Methods section. The details are as follows: 2.12. Statistical Analysis: All experiments included at least three biological replicates. Data are presented as mean ± standard error. Statistical analysis was performed using SPSS 28.0.1.1 and ImageJ. Analysis of variance (ANOVA) was used to compare multiple treatment groups , followed by Tukey’s test (for equal variance) or least significant difference (LSD) test (for unequal variance or more sensitivity) for pairwise comparisons. Statistical significance was set at p ≤ 0.05.
This section is now located on Line 250-256.
Comments 9:
Statistical tests used for comparisons are not described.
Response 9: We agree with this comment and have supplemented the relevant descriptions of statistical tests in the Materials and methods section. This section is now located on Line 250-256.
Comments 10:
Autofluorescence controls using empty vectors were not mentioned.
Response 10: We agree with this comment and have added a description of the empty vector construction method and the corresponding fluorescence results in the Materials and Methods and Results sections. Additionally, we have provided relevant supporting data in the Supplementary Material:
Materials and Methods: “The BiFC plasmids were constructed from the fungal expression vector pIG1783, which carries the A. nidulans gpd promoter, the trpC terminator, and the hph gene for selection of transgenic strains [31]. The plasmids were constructed using the same method as Subotić et al [44]. In the first step, multiple cloning sites (MCS), compatible with the synthesized dsDNA fragments, were enzyme digested with NcoI (NEB) and NotI (NEB), and the green fluorescent protein tag carried by the original plasmid was removed. In the second step, the His-VN173, His-R1-VN173 and His-V1-VN173 fragments for N-terminal labeling of target genes were created by ligating the His-tag, VN173; His-tag, abr1, VN173, and His-tag, abv1, VN173 insert fragments, using primers PIG+His-F and PIG+VN173-R, respectively. In the third step, the HA-VC155, HA-S4-VC155 and HA-Q4-VC155 constructs for C-terminal labeling were generated by synthesizing the corresponding dsDNA fragments, which were then ligated into the vector backbone using primers PIG+HA-F and PIG+VC155-R, respectively (Figure. S1). Finally, the fluorophore sequences were inserted into new vector backbones to yield the final BiFC plasmids: pIG1783-VN173, pIG1783-VC155, pIG1783-R1-VN173, pIG1783-V1-VN173, pIG1783-S4-VC155, and pIG1783-Q4-VC155 (Table S2). Among these, pIG1783-VN173 and pIG1783-VC155 are empty vectors lacking target protein coding sequences and were used as negative controls in the BiFC experiments. The remaining four plasmids contain HD protein coding sequences and were used for functional analysis.”(Line 177-195)
Results: “Strains expressing only the VN173 or VC155 empty vectors did not exhibit yellow fluorescence in the monokaryotic state, nor did they produce fluorescence when mated with each other (Figure S3).” (Line 320-322)
Supplementary Material: “Figure S1-add the fragment PCR gel map used in the empty vectors; Figure S3-add BiFC fluorescence images showing in vivo interaction between VN173 and VC155 empty vectors; Table S2-add empty vector descriptions; Table S3-add primers and fragment sizes used to construct empty vectors.
Comments 11:
Lines 114–117: The description of E. coli expression strains is appropriate, but lacks details such as induction OD600 and expression time.
Response 11: We agree with this comment. We added the following to Materials and Methods: Protein expression was induced at an OD₆₀₀ of 0.4–0.8 using 0.5 mM isopropyl β-D-1-thiogalactopyranoside (IPTG). The cultures were then incubated at 16 °C with shaking at 120 rpm for 16 hours. (Line 129-131)
Comments 12:
Lines 200–207: While microscopy settings are described, the method for fluorescence quantification and normalization (e.g., by area or intensity threshold) should be explained.
Response 12: We agree with this comment. We have reworded this section. The description now reads: The acquired images were processed and analyzed using ZEISS ZEN 3.1 (blue edition) [48], and the fluorescence intensity of the images was measured using imageJ. The size of the measurement area for each image was exactly the same, and the image with the highest fluorescence intensity data was selected for normalization to obtain the relative fluorescence intensity. Raw images were acquired with consistent acquisition settings, including laser power, detector sensitivity, and pixel dwell time, across all samples. (Line 223-229)
Comments 13:
The figures are extremely difficult to interpret due to their small size, low resolution, and the use of excessively small font labels, which significantly hampers readability and data comprehension.
Response 13: We agree with this comment. We have adjusted the font size in Figures 1-3 and adjusted the typography of the images to ensure better visibility and readability.
Figure 1 Line 291
Figure 2 Line 312
Figure 3 Line 334
Comments 14:
BiFC images are presented without quantification of signal intensity (e.g., mean fluorescence in regions of interest with standard error).
Response 14: We agree with this comment. We have added the quantification information of fluorescence intensity of BiFC images in the Supplementary Material: “Figure S4. Relative fluorescence intensity of Figure 3 in the images. Values are presented as mean ± S.E (n = 20). Different letters indicate statistically significant differences (p ≤ 0.05). nd = No fluorescence intensity detected.”
Comments 15:
The pull-down assays and Western blot analyses lack proper negative controls (e.g., a non-interacting protein or untagged version), which would help confirm specificity.
Response 15: We agree with this comment. We have added the relevant content to the manuscript. Materials and Methods: “The Input groups that were not treated with His pull-down were set up as positive controls for the experimental groups, while the samples treated only with HA-tagged proteins were set up as negative controls.” (Line 236-238) Results: “Furthermore, the HA negative controls showed no non-specific bands, indicating the specificity of the interaction assays (Figure 4).” (Line 348-349)
Reviewer 3 Report
Chu et al. studied the role of protein for sexual compatibility and fruiting body development in S. commune and identified interaction of 2 proteins that independently regulate these biological phenomena which hold significance in fungal reproduction and their utility as medicinal source. The introduction is very well written and discusses the background of the mating type, the genotypes and the proteins that play a critical role in fungal reproduction. The experiments are well designed and executed.
N/A
Author Response
Reviewer 3
Comments 1:
Chu et al. studied the role of protein for sexual compatibility and fruiting body development in S. commune and identified interaction of 2 proteins that independently regulate these biological phenomena which hold significance in fungal reproduction and their utility as medicinal source. The introduction is very well written and discusses the background of the mating type, the genotypes and the proteins that play a critical role in fungal reproduction. The experiments are well designed and executed.
Response 1: Thank you very much for your positive comments on our manuscript. We are pleased that you found our introduction well written, experimental design and execution satisfactory. We are also grateful that you recognized our findings on the role of HD proteins in regulating S. commune sexual compatibility and development. Your encouragement affirms the importance of studying these molecular mechanisms, which not only helps to improve our understanding of fungal mating systems, but also helps to explore their potential applications in biotechnology and medicine. Thank you for your valuable comments and we hope you will be our reviewer again for our subsequent studies!